# Marginal Ulcer Perforation after One Anastomosis Gastric Bypass: Surgical Treatment and Two-Year Outcomes

**DOI:** 10.3390/jcm13113075

**Published:** 2024-05-24

**Authors:** Adam Abu-Abeid, Adi Litmanovich, Jonathan Benjamin Yuval, Jawad Tome, Andrei Keidar, Shai Meron Eldar

**Affiliations:** Division of General Surgery, Tel Aviv Sourasky Medical Center, Affiliated to Sackler Faculty of Medicine, Tel Aviv University, 6, Weizman St., Tel Aviv 6423906, Israel; litmana1@biu.ac.il (A.L.); jonathanyu@tlvmc.gov.il (J.B.Y.); jawadt@tlvmc.gov.il (J.T.); andreike@tlvmc.gov.il (A.K.); shaime@tlvmc.gov.il (S.M.E.)

**Keywords:** marginal ulcer, perforation, one anastomosis gastric bypass, complications, surgical revision

## Abstract

**Background:** Marginal ulcer (MU) perforation is a chronic complication after One-anastomosis Gastric Bypass (OAGB). This study’s purpose was to analyze patients undergoing OAGB revision due to MU perforation and describe the two-year outcomes. **Methods:** A retrospective analysis of a database in a single-tertiary hospital. All patients undergoing surgical revision due to MU perforation were included. **Results:** During the study period, 22 patients underwent OAGB revision due to MU perforation. The rate of MU perforation was 0.98%. The median age was 48 years and there were 13 men (59%). The median time from OAGB to MU perforation was 19 months with a median total weight loss of 31.5%. Nine patients (41%) were smokers. Omental patch (±primary closure) was performed in 19 patients (86%) and three patients (14%) underwent conversion to Roux-en-Y gastric bypass (RYGB). At a median follow-up of 48 months, three patients (14%) had recurrent MU diagnosis, of which one had a recurrent MU perforation. Four patients (18%) underwent conversion to RYGB during follow-up. **Conclusions:** MU perforation is a chronic complication after OAGB. In this cohort, most patients were men and likely to be smokers. Omental patch was effective in most cases. Recurrent MU rates at two years follow-up were acceptable.

## 1. Introduction

One Anastomosis Gastric Bypass (OAGB) is the third most commonly performed metabolic and bariatric surgery (MBS) procedure worldwide, comprising 7.6% of MBS [1,2]. Additionally, its prevalence is continuously increasing. OAGB has increased in popularity owing to its relative simplicity, safety profile, efficacy in sustained weight loss and resolution of severe obesity-related diseases [3,4,5].

One of the long-term complications that could be encountered after OAGB is a marginal ulcer (MU) at the gastro–jejunal anastomotic site. MU is reported to occur in 2–3% of patients after OAGB and can present months to years after OAGB [6,7,8]. It is usually associated with heavy smoking and can also be associated with lack of adherence to postoperative prophylactic proton pump inhibitors (PPI), use of non-steroidal anti-inflammatory drugs (NSAIDS), *Helicobacter pylori* (HP) infection, and consumption of corticosteroids [9].

A MU can have several clinical manifestations—in symptomatic cases, it can cause abdominal pain, nausea, vomiting, and dyspepsia, and it can cause strictures due to scar tissue. It can also bleed and present as hematemesis and/or melena. A MU can also perforate and require surgical revision of the OAGB. We have previously published our experience with MU perforations after OAGB [9]. In our previous study, we analyzed 12 patients undergoing surgical revision due to MU perforation after OAGB. We reported the patients’ clinical presentation and surgical treatment. Since then, there were additional patients requiring surgical revision due to MU perforation after OAGB and we could analyze the patients’ follow-up.

The objective of this study was to analyze the two-year outcomes of patients undergoing surgical revision due to MU perforation after OAGB. Our hypothesis was that surgical treatment of MU perforation with omental patching is sufficient in terms of reoperations during the follow up and provides clinical/endoscopic resolution in the majority of cases.

## 2. Materials and Methods

This study is a retrospectively designed study based on a prospectively maintained patient registry database of a single tertiary university hospital. All patients undergoing revisional surgery due to MU perforation after OAGB (January 2015 to December 2023) were included and analyzed in this study.

### 2.1. Patients—OAGB Characteristics

The data regarding OAGB characteristics were captured and included age, sex, baseline body mass index (BMI), smoking, severe obesity related diseases including type 2 diabetes, hypertension, obstructive sleep apnea, hyperlipidemia, and non-alcoholic fatty liver disease. Data regarding previous bariatric procedures, including the procedure type, were also captured. Additional data that were captured, when available, included the biliopancreatic limb length in the OAGB.

### 2.2. Patients—MU Perforation Characteristics

Data captured regarding the MU perforation included the time interval from the OAGB to the MU perforation, the BMI at the time of perforation, the presence of risk factors for MU development, whether a diagnosis of MU was obtained prior to perforation, percentage of total weight loss (%TWL) (calculated as preoperative weight−current weight/preoperative weight×100), and operative findings, such as the size of the perforation, its location (anterior or posterior), and the type of surgical revision performed. Data regarding the postoperative period, including hospital length of stay (LOS), major complications (Clavien–Dindo grade ≥ 3) [10], reoperations and readmissions were captured as well.

### 2.3. Patients—Follow-Up after MU Perforation

Data regarding the follow-up of patients was captured and included, such as the time from the OAGB revision to the last follow up, the last follow up BMI, %TWL, presence of symptoms suggestive of MU [11], and/or upper gastrointestinal (GI) endoscopic findings of MU. We also collected data regarding the need for an additional surgical revision during the follow up. In addition, we performed a univariable analysis to determine risk factors for MU non-resolution at the patients’ last follow up.

### 2.4. Ethical Concerns

This study was approved by the Institutional Review Board (IRB) and was performed in accordance with the ethical standards of the institutional and/or national research committee and with the 1964 Declaration of Helsinki and its later amendments or comparable ethical standards-(TLV-16-0325/11-2019). The IRB of the Tel Aviv Sourasky Medical Center approved a waiver of informed consent for all participating patients due to the retrospective design of the study. All authors declare they have no potential conflicts of interest or financial ties to disclose. No funding was received for this publication.

### 2.5. Statistical Analysis

Statistical analysis was performed with SPSS Statistics (Version 29, SPSS Inc., Armonk, NY, USA). Continuous data are presented as the median (interquartile range (IQR) or mean with standard deviation for data with a normal distribution. A normal distribution was defined as skewness between −3 to 3 and kurtosis −10–10 and all calculations were within this range. Proportions are presented as *n* (%). Dichotomous data were analyzed using the Chi squared or Fisher’s exact test, as appropriate. Continuous data were analyzed using Student’s *t*-test or the Mann–Whitney test, as appropriate.

### 2.6. Surgical Technique and Periopeartive Care—OAGB

All procedures were performed after the administration of venous thromboembolism prophylaxis (subcutaneous injection of heparin 5000 units, 2 h before incision) and antibiotic prophylaxis (intravenous cephazolin 2–3 g, before incision). The patients were placed under general anesthesia, in the supine position and reverse Trendelenburg. The stomach was decompressed with a orogastric tube. The abdomen was insufflated through a Veress needle with carbon dioxide to a pressure of 15 mmHg, and 5 laparoscopic ports were inserted. Dissection of the lesser omentum was initiated using electrocautery until reaching the retro-gastric space (lesser sac). The angle of His was then dissected and the left crus of the diaphragm was cleared. A long and narrow gastric pouch was created below the level of the crow’s foot with a linear stapler against a 36 Fr bougie for calibration. The ligament of Treitz was then identified and the bowel was measured using visual estimation with gradual bowel length measurements of 5 cm. The length of the biliopancreatic limb was measured to 180–200 cm and fashioned according to the patients’ BMI, surgeons‘ preference, and indication for OAGB (primary, or secondary for insufficient outcomes or late complications). A side-to-side gastro–jejunal anastomosis was created between the lateral gastric pouch wall to the small bowel using an endo-stapler-Echelon blue cartridge (Ethicon Endo-Surgery Inc., Cincinnati, OH, USA) or End-GIA purple cartridge (Covidien/Medtronic Inc., Minneapolis, MN, USA). The anastomotic opening was then sutured using a barbed suture. A routine blue dye patency and leak test was then performed.

All patients were admitted to the surgical ward following surgery. Instructions for high dose PPI (40 mg bid) and venous thromboprophylaxis were given to all patients. The patients were instructed on early mobilization on the day of the surgery and on postoperative day one gradually resumed an oral diet, starting with a liquid diet. Patients were considered dischargeable when they had the ability to tolerate fluids, a liquid diet was consumed, and when ambulation was fully resumed. Upon discharge, all patients received a prescription for high-dose proton pump inhibitors (40 mg bid) for three to six months, subcutaneous low molecular weight heparin injections for three weeks, and a strict recommendation for oral multi-vitamin supplements lifelong. Smokers were strongly encouraged to cease smoking. All patients were scheduled for a routine follow-up two weeks postoperatively, followed by a 1-, 3-, 6-, one year and then a yearly follow-up. During the follow-ups, all patients were assessed on the outcomes of MBS including, weight loss, resolution of severe obesity related conditions, 30 day complications, and chronic complications including MU, and were screened for nutritional deficiencies.

### 2.7. Preoperative Management of MU Perforation

Following a MU perforation diagnosis, broad-spectrum intravenous antibiotics (including anti-fungal coverage) administration and intravenous fluid resuscitation was initiated. All patients were given intravenous pantoprazole 80 mg, followed by continuous intravenous pantoprazole administration. A nasogastric tube and urinary Foley catheter were inserted prior to surgery.

### 2.8. Surgical Techniques—Revisional Surgery for MU Perforation

All procedures were performed in the laparoscopic approach (with conversion to open surgery when indicated) and patients were in the supine French position. The abdomen was explored, and a diagnosis of MU perforation was made. In cases with purulent peritonitis, a thorough abdominal washout was performed. The operative procedures are described below:Laparoscopic omental patch with or without primary repair: The MU perforation was identified and assessed if suitable for primary repair. In cases of primary repair, suturing was conducted using interrupted absorbable sutures. At least three sutures were placed perpendicular to the perforated MU and a patch of omentum was then brought beneath the sutures, which were then tied. A routine blue dye leak test was then performed. Routinely, a drain was placed along the aspect of the MU repair and omental patch.Conversion to Roux-en-Y Gastric Bypass (RYGB): The MU perforation was identified, and the gastro–jejunal anastomosis was resected (which included the MU) with linear staplers. A stapled jejuno–jejunal anastomosis was then performed and the anastomotic opening was closed with a barbed suture. The ligament of Treitz was then identified and the bowel was transected 50–100 cm distally, defining the length of the biliopancreatic limb. A stapled gastro–jejunal anastomosis was then performed between the gastric pouch and the Roux limb with closure of the opening using a barbed suture. The bowel was then measured 150 cm distally to define the length of the Roux limb and a jejuno–jejunal anastomosis was performed using a linear stapler with manual suturing of the opening with a barbed suture. The mesenteric defects of the jejuno–jejunal anastomoses and the Petersen defect were closed using non-absorbable interrupted sutures. A routine blue dye leak test for the gastro–jejunal anastomosis was then performed. A drain was placed routinely along the gastro–jejunal anastomosis.

### 2.9. Postoperative Care

All patients were admitted to the surgery ward, were usually continued on high-dose intravenous PPI (40 mg bid) and intravenous antibiotics including antifungal treatment for four days. Gradually, the nasogastric tube was removed, and these patients gradually resumed a liquid diet and were discharged once ambulating freely, afebrile without antibiotic treatment, appropriate analgesia was achieved with oral non-narcotic medications and when a liquid diet was tolerated. At discharge, all patients were prescribed oral high dose PPI (40 mg bid) and were recommended to undergo an upper GI endoscopy six weeks following surgical revision to ensure MU healing.

## 3. Results

During the corresponding period, 22 patients underwent surgical revision due to a MU perforation after OAGB, among which 15 patients underwent OAGB in our center, which accounts for a 0.98% rate of MU perforation (15/1522). The baseline characteristics of the patients prior to OAGB are shown in Table 1—the median age was 48.6 (IQR—23.4), there were 13 men (59%), and the median BMI prior to OAGB was 38 (IQR—6.9). The median biliopancreatic limb length was 195 cm. There were 11 patients (50%) who were smokers, 5 patients (23%) who had a preoperative HP eradication (the rest were negative), and 8 patients (36%) who underwent a previous bariatric procedure including sleeve gastrectomy (*n* = 5) and laparoscopic adjustable gastric band (*n* = 3).

### 3.1. Characteristics of MU Perforation

The characteristics of patients with a MU perforation are shown in Table 2. The median time from OAGB to MU perforation was 19 months (IQR—18), and seven patients (32%) had a preoperative diagnosis of MU. The median BMI and TWL at MU perforation were 24.5 kg/m^2^ and 31.5%, respectively. When considering risk factors for MU perforation, 9/22 (41%) were smokers, 3/22 (14%) were taking NSAIDS, 2/22 (9%) were taking corticosteroids, and one patient had a known HP infection (5%). In addition, 13/22 patients (59%) complied with the recommendation of 3–6 months of prophylactic PPI consumption. The median hospital length of stay was seven days and none of the patients had a major complication (Clavien–Dindo grade ≥ 3). The operative procedures performed are shown in Figure 1. The median MU perforation size was 5 mm and in 18/22 (82%) patients it was located in an anterior aspect. Nineteen patients (86%) underwent omental patch with or without primary repair and three patients (14%) underwent conversion to RYGB.

### 3.2. Two-Year Follow-Up

The long-term follow-up data are shown in Table 3. All patients were available for follow-up, with a median follow up of 48 months (IQR 23.7) from OAGB and 28 months (IQR 17.5) from the time of MU perforation. The median BMI and %TWL at the last follow-up was 24.3 kg/m^2^ and 35.7%. Most (20/22, 91%) patients were still on PPI treatment. Patients with recurrence/non-healing of the MU and patients requiring additional revisional surgery are described below:(1)*Patients with recurrent MU diagnosis on upper GI endoscopy*

There were three (14%) patients with a recurrent MU diagnosis after omental patching. One patient did not comply with postoperative PPI consumption, one patient’s MU did not heal as noted during endoscopic surveillance and underwent conversion to RYGB, and the third patient had an MU on upper GI endoscopy follow-up that was treated endoscopically with a jejunal flap; despite that, the MU recurred and the patient eventually underwent conversion to RYGB with MU resolution.

(2)
*Patients with recurrent MU perforation*


One patient (5%) had a recurrent MU perforation. This 67-year-old patient underwent conversion from sleeve gastrectomy to OAGB due to sleeve stenosis. Four months after the OAGB, she suffered from a MU perforation and underwent laparoscopic omental patching. Following that, there was complete clinical and endoscopic resolution of the MU. Later on, during the follow-up, the patient suffered from severe protein energy malnutrition, which required recurrent admissions and intravenous nutritional care. The patient was reluctant to undergo additional revisional surgery. Three years later, the patient presented with a recurrent MU perforation, which necessitated an emergent surgical exploration and was treated with omental patching due to poor nutritional status and severe septic shock during surgery.

(3)
*Patients converted to RYGB*


Two patients (9%) required conversion to RYGB due to recurrent MU, and one patient (5%) underwent conversion to RYGB due to biliary reflux, which did not improve with conservative means (lifestyle modifications and PPI treatment).

Of the 19 patients treated initially by omental patching, 15 (79% of these patients and 68% of the whole cohort) had complete resolution of the MU during follow-up.

### 3.3. Risk Factors for Non-Resolution of MU

We performed a univariable analysis of baseline and MU perforation patient characteristics between patients with MU resolution at follow-up (*n* = 18) to patients with recurrent disease (*n* = 4). No meaningful difference was seen between the groups. The variables that were closest to reaching statistically significant differences between patients with and without MU resolution were pre-OAGB BMI (40.3 vs. 32.8; *p* = 0.076), biliopancreatic limb length (193.3 cm vs. 153.8 cm; *p* = 0.086), age at MU perforation (41.3 years vs. 54.4 years; *p* = 0.091) and LOS at MU perforation (8.2 days vs. 15.5 days; *p* = 0.068).

## 4. Discussion

In this case series we showed the clinical course and management of MU in patients following OAGB, including 2-year weight loss outcomes. MU was diagnosed at a median of one and a half years following OAGB and patients with MU were predominantly men. The majority of MU perforations were located in an anterior aspect. Most patients were treated by omental patch (with or without primary repair) and this treatment was both safe and efficacious in the majority of cases, although four patients (~20%) eventually required elective conversion to RYGB during follow-up and only three (14%) were due to recurrent or unresolved MU. Given the relatively small sample size, the proportion of revisions necessary following omental patching may be somewhat inaccurate. Weight loss outcomes of patients experiencing MU perforation were not meaningfully different from historical controls not experiencing MU perforation [12,13].

OAGB has been suggested to be an ulcerogenic procedure due to the long acid producing gastric pouch, bile reflux and increased tension on the anastomosis due to a long Billroth 2 constructed biliopancreatic limb [8]. The incidence of marginal ulcers after OAGB varies in different studies. Braga and colleagues performed an endoscopy on 39 OAGB patients 24 months post-operatively [14]. They reported four cases of MU representing 10.3%. In a recent systematic review and meta-analysis, the reported prevalence was 2.6% [7]. In an international survey of MBS surgeons that performed a combined total of 27,672 OAGB, the reported MU rate was 2.2% [6]. We were not able to report the MU rate since not all patients underwent upper GI endoscopy; however, the prevalence of MU perforations was 15/1522 OAGBs (0.98%), which is comparable to previous reports after RYGB of 0.47–0.82% [15,16].

MU has several known risk factors. We found that the majority of patients presenting with perforated MU were men (13/22, 59%) although women make up the vast majority of patients undergoing OAGB. This is in contrast to a systematic review published by Martinino and colleagues on perforated marginal ulcer following gastric bypass with 67% of reported patients being women [17]. Their study was based primarily on RYGB, and there are multiple additional and cultural factors that may play a role in the pathophysiology and development of MU perforation. The finding of a male predominance could be related to the fact that male patients have a greater tendency to smoke [18], which is a well-documented risk factor for MU [19], and men are also reported to be less adherent to post-operative clinical follow up [20]. Our findings support cigarette smoking as a risk factor for the development of MU and its perforation, as nine of our patients were active smokers (41%). Possible mechanisms leading to MU formation in smokers are decreased blood flow to the mucosa and alterations in the mucosal immune system [21]. In our opinion, when choosing OAGB, smoking habits should be taken into consideration as it is a major risk factor for the development of MU and perforation. Smokers should be strongly encouraged to quit smoking prior to OAGB. Smokers with a chronic ulcer who are unable to cease smoking and develop an ulcer perforation should be considered for OAGB reversal to normal anatomy, as maintaining a gastro–jejunal anastomosis while they continue smoking puts them at risk for recurrent ulcer and recurrent perforation. 

Herein, we describe the characteristics of perforation as well as the operative and post-operative course, including recurrence and re-operation. Our patients presented with MU perforation at an average of 19 months following their OAGB. This is in line with reports on MU in general, with 50% reported as late MUs at least 12 months following gastric bypass surgery [22]. In our series, 19 of the 22 (86%) patients underwent omentopexy with or without primary suturing of the perforation. Only three patients (14%), all of which were known to have a chronic non-healing ulcer prior to their current presentation, underwent resection of their anastomosis and conversion to RYGB at presentation, with an additional four patients (18%) undergoing conversion during follow-up. 

The optimal surgical treatment in the emergency setting of a perforated MU is debatable. Omentopexy with or without primary suture of the perforation is a reasonable solution, with acceptable outcomes as shown herein. Surgical revision due to MU perforation depends on several factors such as the MU position, the size of the perforation, the patient’s clinical status during the surgery, and the surgeon’s experience [9]. We recommend involving experienced MBS surgeons early in the workup and management of patients with MU perforations for optimal outcomes. Some authors have urged performing diversion of the biliary flow away from the gastro–jejunal anastomosis whether by conversion to RYGB or by adding a Braun anastomosis to the omental patch [23]. In fact, Felsenreich and colleagues showed overt bile reflux in endoscopy five years following OAGB in a significant minority of patients (43%) [24]. 

In our series, acceptable perioperative and 2-year outcomes were seen in patients following omental patching without the diversion of bile. It seems that bile diversion should be performed on a case-by-case basis. At a median follow-up of two years following MU perforation, four (18%) patients had MU recurrence, one of which suffered from recurrent MU perforation. To our knowledge this is the first report of the rate of recurrent MU following perforation of an MU after OAGB. In line with our findings, Moon and colleagues showed that in a retrospective case series on MU perforations in RYGB, 8/12 patients (67%) were treated with omental patch and at least two patients (17%) had a recurrence at a mean follow-up of 28.9 months [16]. This may call into question the higher ulcerogenic potential of OAGB in comparison to RYGB.

Our study has several limitations, which include the inherent biases of a small sample size and the study’s retrospective design. In addition, not all patients after OAGB were available for follow-up and we could not accurately estimate the rate of MU after OAGB. Despite these limitations, we provide some important insights into the clinical course and management of patients with MU perforation following OAGB. The strengths of this study include the granularity of the data and relatively long follow-up. In addition, this is, to our knowledge, the largest case series reported on perforated marginal ulcers following OAGB.

## 5. Conclusions

In conclusion, MU perforations were not shown to be extremely rare in our experience (0.98%). In our series, most patients were male, experienced an anterior perforation, and were treated effectively with an omental patch. Recurrent MU rates at 2-year follow-up were reasonable and in the range of acceptable rates. Further large-scale and prospective studies are needed to define the prevalence of MU in OAGB and to compare the prevalence of MU between different MBS procedures (especially RYGB).

## Figures and Tables

**Figure 1 jcm-13-03075-f001:**
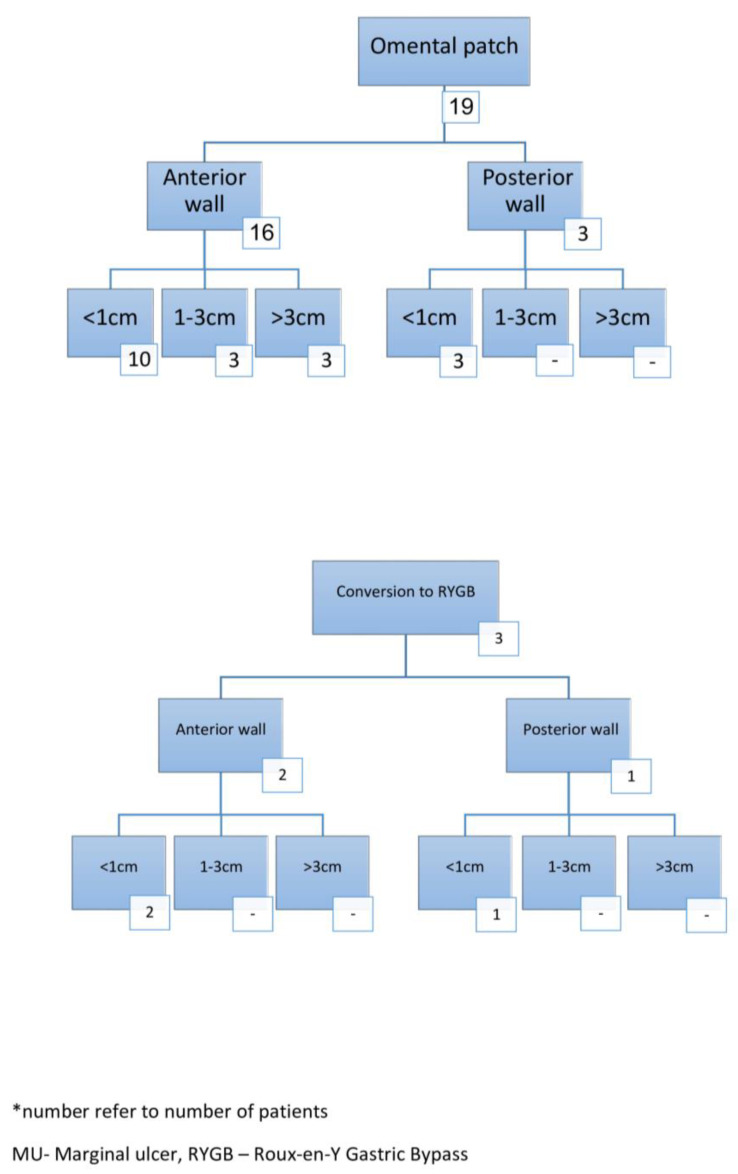
Operative findings in patients with MU perforation *.

**Table 1 jcm-13-03075-t001:** Baseline characteristics at OAGB.

Age at OAGB median (IQR), years	48.6 (23.4)	40.70 (15.1) *
Sex, female:male	9:13	
BMI before OAGB, median (IQR), kg/m^2^	38.0 (6.9)	37.92 (7.4) *
Smoking before OAGB, *n*	11/22	
Diabetes mellitus, *n*	6/22	
OSA, *n*	8/22	
Hypertension, *n*	12/22	
Hyperlipidemia, *n*	10/22	
NAFLD, *n*	17/22	
Preoperative HP eradication, *n*	5/22	
Previous bariatric surgery, *n*	8/22	
BPL length, median (IQR), cm	195 (20)	176.6 (38.8) *

OAGB—One Anastomosis Gastric Bypass, BMI—Body mass index, IQR—Interquartile range, OSA—Obstructive sleep apnea, NAFLD—Non-alcoholic fatty liver disease, HP—*Helicobacter pylori*, BPL—Biliopancreatic limb. * refers to mean (standard deviation).

**Table 2 jcm-13-03075-t002:** Characteristics of patients with MU perforation following OAGB and perioperative data at perforation.

Age at perforation, median (IQR) years	49.9 (21.7)	42.6 (14.7) *
BMI at perf. median (IQR), kg/m^2^	24.5 (8.3)	24.9 (7) *
%TWL, median (IQR)	31.5 (24.6)	
Time to perforation median (IQR), Months	19 (18)	15.1 (15.5) *
Known MU prior to perf. *n*	7/22	
Risk factors, *n*		
PPI use between OAGB and perforation	13/22	
Smoking	9/22	
NSAIDS	3/22	
Steroids	2/22	
HP infection	1/22	
Early postoperative outcomes		
LOS, median (IQR), days	7 (4.5)	6.2 (11.4) *
CD ≥ 3, *n*	0/22	

BMI—Body mass index, IQR—Interquartile range, MU—Marginal ulcer, PPI—Proton pump inhibitor, OAGB—One Anastomosis Gastric Bypass, HP—*Helicobacter pylori*, LOS—Length of stay, CD—Clavien–Dindo. * refers to mean (standard deviation).

**Table 3 jcm-13-03075-t003:** Mid-term follow-up of patients with MU perforation.

Follow-up time		
from OAGB, median (IQR), months	48 (23.75)	49.2 (21.2) *
from perforation, median (IQR), months	28 (17.5)	25.3 (20.9) *
BMI, median (IQR), kg/m^2^	24.3 (8.7)	26.1 (6.9) *
%TWL, median (IQR)	35.7 (27.2)	
PPI use, *n*	20/22	
Additional revisional surgery	2/22	
MU in endoscopy, *n*	3/22	
Symptoms suggestive of MU, *n*	4/22	
MU re-perforation, *n*	1/22	

MU—Marginal ulcer, OAGB—One Anastomosis Gastric Bypass, BMI—Body mass index, TWL—Total weight loss, PPI—Proton pump inhibitors. * refers to mean (standard deviation).

## Data Availability

The raw data supporting the conclusions of this article will be made available by the authors on request.

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
