# Peer review of "Marginal Ulcer Perforation after One Anastomosis Gastric Bypass: Surgical Treatment and Two-Year Outcomes"

_jcm, 2024, doi:10.3390/jcm13113075_

Round 1

Reviewer 1 Report

Comments and Suggestions for Authors

Thanks for the Editors for the opportunity to review this paper. It is a well-written manuscript about an important complication following a bariatric procedure that is increasingly performed.

Introduction: OAGB is performed in 7.6% of cases according to IFSO data in 2021. (ap. 4% only for MENAC)

We use the term 'obesity related-diseases'.

Materials and methods: Write the number and name of IRB agreement.

Name the statistics program. How did you check the normality of distribution? Did all data require IQR not SD?

How the anastomosis was performed, on which stomach wall. What stapler was used? What thread was sewn with, if so? What bougie size you use? What was the lenght of the bp limb?

What were 'high doses of PPIs'?

Results: what primary surgery have the patients undergone before OAGB?

Only 5 patients/22 had HP eradication? Were the rest negative?

Did the authors consider performing additional statistical tests to demonstrate the OR of perforation, e.g. uni and then multivariate logistic regression?

Conclusions: extract conclusions as a new paragraph.

"In conclusion MU perforations are not extremely rare (%)" What do you mean? 

Author Response

1.Introduction: OAGB is performed in 7.6% of cases according to IFSO data in 2021. (ap. 4% only for MENAC)

Response – We thank the reviewer for pointing this and we have corrected this mistake. Lines -29-30

2.We use the term 'obesity related-diseases'.

Response – We have rephrased this term throughout the manuscript to obesity related diseases as suggested by the reviewer.

3.Materials and methods: Write the number and name of IRB agreement.

Response – We added this data to the revised manuscript.

4.Name the statistics program. How did you check the normality of distribution? Did all data require IQR not SD?

Response - We used SPSS Statistics version 29, we added this to the revised manuscript. Normal distribution was defined as Skewness between -3 to 3 and Kurtosis -10-10 and all calculations were within this range.  In accordance with the reviewr’s comment we added to the tables the mean and standard deviation for variables with normal distribution.

5.How the anastomosis was performed, on which stomach wall. What stapler was used? What thread was sewn with, if so? What bougie size you use? What was the lenght of the bp limb?

Response – We thank the reviewer for noting this and we have clarified all this data to the revised manuscript – Lines 107-117

6.What were 'high doses of PPIs'?

Response: We have written that patients are given 40mg bid for 3-6 mpnths after surgery. After MU perforation diagnosis all patients are given IV 80mg Pantoprazole followed by a continous infusion.

7.Results: what primary surgery have the patients undergone before OAGB?

Response: We have added the primary surgeries that were performed to the revised manuscript which included – 8 patients (36%) who underwent a previous bariatric procedure including sleeve gastrectomy (n=5) and laparoscopic adjustable gastric band (n=3) – Lines – 182-183

8.Only 5 patients/22 had HP eradication? Were the rest negative?

Response: 5/22 patients had HP infection, the rest were negative, we have clarified this in the revised manuscript – line 181

9.Did the authors consider performing additional statistical tests to demonstrate the OR of perforation, e.g. uni and then multivariate logistic regression?

Response: We thank the reviewer for the insightful comment. We did not perform a statistical analysis to demonstrate the OR of perforation as we did not have a control group for this cohort. Following the reviewer’s valuable comment we have decided to perform a univariable analysis of risk factors for MU non-resolution at follow-up. Because no variable was statistically significant on a univarable analysis and the number of events was small (n=4) we did not perform a multivariable analysis. We added this data to the revised manuscript

10.Conclusions: extract conclusions as a new paragraph.

Response: In accordance with the reviewer’s comment we have extracted the conclusions as a new paragraph

11."In conclusion MU perforations are not extremely rare (%)" What do you mean?

Response: We mistakenly did not write the percentage of MU perforation in our experience. We have rephrased this sentence - In conclusion, MU perforations were not shown to be extremely rare in our experience (0.98%). We thank the reviewer for pointing this error.

Reviewer 2 Report

Comments and Suggestions for Authors

This is an interesting and well-written paper. The English is excellent.

It does have some limitations, for example it is a small and retrospective review. However, the review is well written and gives useful data.

The findings are not surprising but it is a useful review, which addresses a common problem. The preponderence of male patients and smokers are not surprising and probably linked but the authors do point this out. 

There is a typo in Section 3.1. It should be " Clavien-Dindo". I could find no other mistakes.

The discussion is interesting, well-written and of an appropriate length.

Overall, it is an interesting paper, which addresses a common clinical problem and gives useful discussion of the treatment options.

Author Response

Reviewer 2

This is an interesting and well-written paper. The English is excellent. 

It does have some limitations, for example it is a small and retrospective review. However, the review is well written and gives useful data.

The findings are not surprising but it is a useful review, which addresses a common problem. The preponderence of male patients and smokers are not surprising and probably linked but the authors do point this out. 

There is a typo in Section 3.1. It should be " Clavien-Dindo". I could find no other mistakes.

The discussion is interesting, well-written and of an appropriate length.

 Overall, it is an interesting paper, which addresses a common clinical problem and gives useful discussion of the treatment options.

Response: We thank the reviewer for the positive comments, we have corrected the typo accordingly – section 3.1

Round 2

Reviewer 1 Report

Comments and Suggestions for Authors

The authors did a good work to improve the manuscript. I suggest to accept the paper